# Scaling Manipulation Learning with Visual Kinematic Chain Prediction

**Xinyu Zhang**    **Yuhan Liu**    **Haonan Chang**    **Abdeslam Boularias**
{xz653, yl1834, hc856, ab1544}@rutgers.edu
Rutgers University

**Abstract:** Learning general-purpose models from diverse datasets has achieved great success in machine learning. In robotics, however, existing methods in multi-task learning are typically constrained to a single robot and workspace, while recent work such as RT-X requires a non-trivial action normalization procedure to manually bridge the gap between different action spaces in diverse environments. In this paper, we propose the visual kinematics chain as a precise and universal representation of quasi-static actions for robot learning over diverse environments, which requires no manual adjustment since the visual kinematic chains can be automatically obtained from the robot's model and camera parameters. We propose the Visual Kinematics Transformer (VKT), a convolution-free architecture that supports an arbitrary number of camera viewpoints, and that is trained with a single objective of forecasting kinematic structures through optimal point-set matching. We demonstrate the superior performance of VKT over BC transformers as a general agent on Calvin, RLBench, ALOHA, Open-X, and real robot manipulation tasks. Video demonstrations and source code can be found at https://mlzxy.github.io/visual-kinetic-chain.

**Keywords:** Multi-Task Robot Learning, Manipulation

## 1 Introduction

There are numerous techniques in machine learning and computer vision that can successfully learn a single general-purpose model from multiple diverse datasets [1]. In robotics, however, despite the recent advances in multi-task learning that enable a single policy to perform various tasks through imitation learning with language instructions [2, 3], these methods are typically constrained to a single robot and workspace. Some recent work, such as RT-X [4, 5], leverages a large vision-language model (VLM) to directly train policies with Behavioral Cloning (BC) on the Open-X Embodiment [5], a collection of datasets crowd-sourced from various environments and robots. However, these techniques require a non-trivial action normalization procedure to bridge the gap between the different action spaces in diverse setups, such as end-effector poses, joint positions, and velocities in various world frames. This manual engineering procedure is currently custom-designed for each training dataset, which affects the generalization and interpretability of these models.

Therefore, a key question is: can we find an action representation that is precise and universal for various setups and robots? To this end, we propose the *visual kinematic chain*, which is the projection of the robot's high-dimensional kinematic structure into the image plane as pixels. Instead of predicting the low-level robot actions, we propose to learn and visually forecast the movement of robots' kinematic chains in the image plane. This visual approach provides a unified action representation for different robots, and requires no manual adjustment since the visual kinematic chains can be automatically obtained from the robot's model and camera parameters.

To forecast the kinematic structures of various robots, we render the kinematic chains into point sets by sampling points along links and performing optimal matching [6] to minimize the earth moving

8th Conference on Robot Learning (CoRL 2024), Munich, Germany.

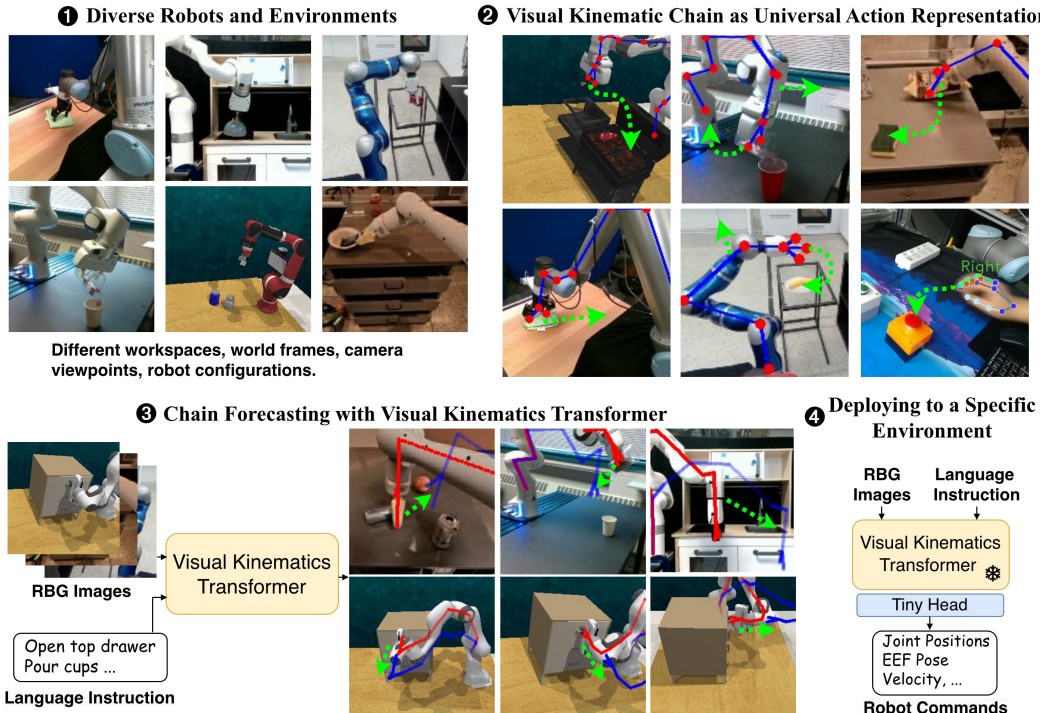

Figure 1: **Overview of the proposed framework.** We use the visual kinematic chain as a universal action representation across diverse robots and setups. We propose the Visual Kinematics Transformer (VKT), an architecture based solely on attention layers, which predicts the future movements of the visual kinematic chain in images from multiple viewpoints. Our VKT is trained with the earth-moving distance as the single learning objective without knowing any low-level robot states or actions. When deployed in a specific environment, we freeze the VKT as a backbone and attach a tiny head to project the VKT output to actual robot commands.

distance [7, 8] between the predicted point sets and ground-truth kinematic structures. Further, we propose the **Visual Kinematics Transformer** (**VKT**), a convolution-free architecture that is solely based on the attention mechanism in the RGB space, which supports an arbitrary number of camera viewpoints. Our VKT is trained with a single objective of forecasting kinematic structures through optimal point set matching without seeing any low-level robot actions. VKT is deployed to a specific environment by simply training a tiny head while freezing the backbone. VKT demonstrates superior performance over BC transformers as a general agent on Calvin (average length 1.46 vs 0.48), RLBench (success rate 61.7% vs 24.3%), ALOHA (success rate 55.3% vs 12%), Open-X, and real robot manipulation tasks.

To summarize, our contributions are threefold. (1) We propose the visual kinematic chain as a precise and universal representation of quasi-static actions for learning from diverse robot configurations. (2) We propose VKT, a convolution-free architecture that supports an arbitrary number of camera viewpoints and is trained with a single objective of forecasting kinematic structures through optimal point set matching. (3) We present a thorough empirical study of the performance of VKT on specialized and general agents over a diverse set of language-conditioned tasks and environments.

## 2 Related Work

**Robot Learning from Diverse Environments.** Learning a single universal multi-task policy on various robots, cameras, and task configurations is a challenging problem that is receiving increasing attention [9]. One recent strategy to solving this problem consists in training a network that takes as input a robot's kinematic structure [10, 11]. For instance, MetaMorph [12] tokenizes the

robot's kinematic tree, which is then used as a transformer prompt to produce per-joint action. However, these methods are only evaluated in simple walking environments of Mujoco [13] without any variations in the task, the camera viewpoint, and the workspace. Several recent works have also looked into leveraging a large vision-language model (VLM) to directly train robots, through behavior cloning, on a collection of crowd-sourced datasets such as Open-X Embodiment [14, 15, 5]. Despite their impressive results, these methods require a non-trivial action normalization procedure to reduce the gap between action spaces such as end-effector poses, joint positions, and velocities in various world frames. However, this manual engineering procedure is custom-designed for each training dataset, which severely affects the generalization and interpretability of these models. In contrast, our method directly learns and visually predicts the movement of robots' kinematic chains.

**Transformers for Manipulation Learning.** Transformers are widely used in manipulation learning [4, 16]. However, existing architectures require a fixed number of camera viewpoints, which is not guaranteed when learning is performed across various environments. Both RT-1/2 and RT-X select one canonical view from each Open-X dataset [5, 4, 17]. RVT [18] supports multiple but predefined viewpoints. PerAct [3] trains transformers over point-clouds and is not limited by camera viewpoints. But methods based on point clouds still suffer from the lack of datasets with depth information, as well as scale discrepancies [19, 20]. In contrast, our visual kinematics transformer (VKT) is a convolution-free architecture that is solely based on attention mechanisms in the RGB space, without the need for depth data. VKT also supports an arbitrary number of camera viewpoints.

**Intermediate Action Representations.** Instead of only using low-level robot commands, incorporating intermediate actions into control policies has been shown to increase data efficiency [21]. RT-H is a hierarchical policy that predicts language motions such as "move arm forward" [22]. SWIM uses affordance maps to unify grasping actions of humans and robots [21]. General Flow employs a labeling process to translate end-effector motions into point-cloud movements [23]. IMOP [24] uses invariant regions to describe key manipulation poses. RT-Trajectory predicts the future 2D trajectory of the end-effector's center in a given image [25]. In contrast, our method forecasts the movements of the entire kinematic chain in multiple camera viewpoints. This representation provides a visually grounded action abstraction, while capturing detailed and small motions at the same time.

Recently, designing universal action representation has received growing attention. ATM [26] designs a track-guided policy that accepts a set of point trajectories on the image plane as guidance for predicting robot actions, which enables policy learning from both robot and human videos. However, compared to our visual kinematic chain, point trajectory is ambiguous because trajectories are not guaranteed to correlate with the desired robot actions. UniPi [27] solves robot tasks by first generating videos and then deriving robot actions from the video generated. Yet, using videos as an action representation is inefficient because video generators are orders larger than common policy networks [28, 29]. Moreover, UniPi [27] is evaluated on a single planar environment [30] because deriving 3D actions or joint positions from videos is not trivial.

## 3 Visual Kinematic Chain Forecasting

We consider the problem of learning a single universal multi-task policy from diverse robots and workspaces through a unified action space. Our key insight lies in the use of the *visual kinematic chain*, which is the projection of the robot's high-dimensional kinematic structure into a set of pixels in the image 2D plane. The visual kinematic chain can be analytically generated with camera parameters and robot models without any manual engineering.

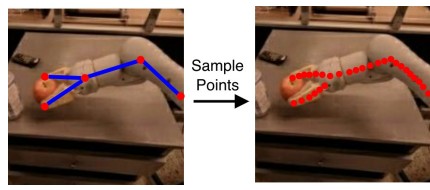

Figure 2: Rendering a kinematic chain as a set of pixels in an RGB image

We show that this simple representation allows us to train a single visual language-conditioned policy over multiple environments. The VLM policy is trained with a single objective of forecasting the future visual kinematic chain movements without knowing any low-level robot actions. Our proposed transformer architecture is built entirely with attention layers and can predict consistent

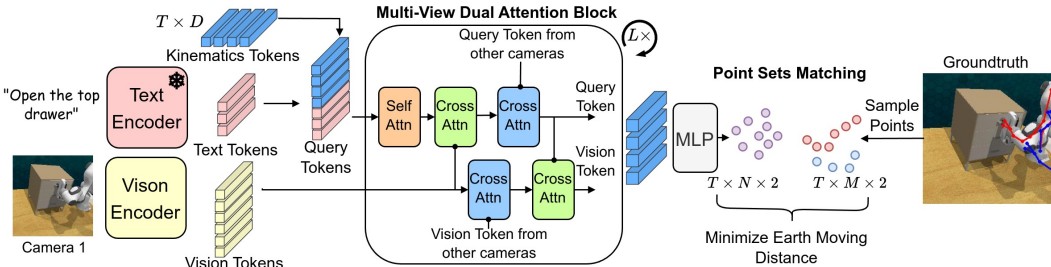

Figure 3: **Overview of our proposed visual kinematics transformer (VKT).** For each camera input, we encode the language instruction and RGB image as text-and-vision tokens with CLIP [32]. Then, we concatenate the text tokens and the kinematics tokens as query tokens. The kinematics tokens are learned parameters. Next, the query and vision tokens are interweaved with a sequence of our proposed multi-view dual attention block. For each block, the query tokens are first updated with self-attention (orange). Then, cross attention is applied with the query tokens as queries and the vision tokens as keys and values (green). A cross-attention layer updates queries with keys and values, we use → to denote queries and ─• to denote keys and values. Then, both the query tokens and the vision tokens are updated through cross-attentions with tokens from other camera viewpoints (blue). Next, the vision tokens are updated by cross-attention with query tokens as keys and values (green). Finally, the $T$ kinematics tokens are projected into $T$ point sets through an MLP, representing the visual kinematic chain in the current and the future $T - 1$ steps. The predicted point-sets are optimized through point-set matching with the ground-truth, as shown in Equation 1.

multi-view chains from an arbitrary number of camera viewpoints. To deploy the multi-task policy in a specific environment, one simply needs to freeze the transformer and train a tiny head to project the actions into actual robot commands. An overview of our framework is illustrated in Figure 1.

**Fitting any Structure with Point-Set Matching.** The kinematic structures of various robots differ significantly. To forecast different structures, we propose to render them to point sets by uniformly sampling points along links, as illustrated in Figure 2. Let $P = \{p_i\}_{i=1}^M$ denote the point-set rendered from the ground-truth configuration of a kinematic chain, and $Q = \{q_i\}_{i=1}^N$ denote the point-set predicted by VKT, where $p_i \in \mathbb{R}^2, q_i \in \mathbb{R}^2$. We first compute the pairwise euclidean distance matrix $C \in \mathbb{R}^{M \times N}$, where $C_{ij} = \| p_i - q_j \|$. We then minimize the earth-moving distance $\text{EMD}(P, Q) = \sum_{i,j} \gamma_{ij}^* C_{ij}$, where $\gamma^*$ is a stochastic assignment matrix that matches points in $P$ and $Q$. We find $\gamma^*$ using the Sinkhorn-Knopp algorithm to solve the following optimization problem online for each mini-batch [31, 8].

$$\gamma^* = \arg\min_{\gamma \in \mathbb{R}_+^{M \times N}} \sum_{i,j} \gamma_{ij} C_{ij}, \text{ s.t. } \gamma 1 = 1; \gamma^T 1 = 1; \gamma \geq 0. \tag{1}$$

**Visual Kinematics Transformer (VKT).** Our VKT network accepts a language instruction and an RGB image as input and predicts a sequence of $T$ point-sets. The dimensions of the kinematics tokens are $T \times D$, where $D$ denotes the embedding size, and each token predicts one point-set. The first point-set in the predicted sequence describes the current state of the kinematic chain in the given RGB image. The remaining point-sets are predictions of the future states of the kinematic chain in the next $T - 1$ time-steps, i.e., future robot movements. The network is trained to match each point-set to the ground-truth kinematic structure at the corresponding time-step by minimizing the earth-moving distance. The same process is applied to multiple RGB images taken from different viewpoints. The network is also trained to make the future point-sets predicted from different viewpoints consistent with each other. The kinematics tokens are concatenated with the text tokens as query tokens. The concatenated query and vision tokens are forwarded to our proposed *multi-view dual attention block*, to interweave the information between query and vision tokens and across multiple camera viewpoints.

The multi-view dual attention block performs dual attentions between query and vision tokens within each single view, indicated by the green blocks in Figure 3, and multi-view attentions for query

and vision tokens independently but across multiple camera viewpoints, shown as the blue blocks in Figure 3. This dual attention enables the query token to learn the visual kinematic chain movements. The multi-view attention enables a spatially-consistent prediction across multiple viewpoints. The output kinematics tokens are projected to $T$ point-sets using a small MLP network. Each point-set contains $N$ 2D points on the input image. The VKT architecture is illustrated in Figure 3.

Compared to existing robot learning transformers [18, 17], our proposed VKT is convolution-free and built solely with attention layers. Therefore, VKT supports an arbitrary number of camera viewpoints, which is an important advantage as camera setups are different among various environments.

**Deploying to a Specific Environment.** We freeze the trained VKT, drop the point-set prediction branch, and only use the kinematics tokens. We apply a 1D convolution to project the $T$ kinematics tokens into low-level robot actions for the next $T$ timesteps. Unlike the VKT backbone which is trained from multiple environments, the 1D convolution head is trained separately for each environment. The corresponding proposed architecture is illustrated in Figure 4.

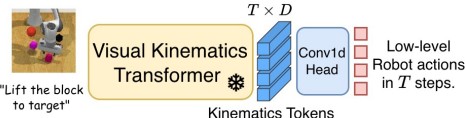

Figure 4: Projecting the kinematics tokens into robot actions with 1D convolution.

Table 1: **Comparison of Behavioral Cloning Transformer (BCT) and our Visual Kinematics Transformer (VKT) on Calvin, RLBench, and ALOHA.** BCT and VKT share the same architecture except that BCT directly predicts the robot's low-level actions. VKT outperforms BCT in isolated environments (specialized). When trained in all environments (general), VKT retains a competitive performance but BCT suffers a drastic performance drop.

| Environment | Metric | Specialized Agent | | General Agent | |
|---|---|---|---|---|---|
| | | BCT | VKT (Ours) | BCT | VKT (Ours) |
| Calvin [33] | Avg. Length ↑ | 1.36 | **1.58** | 0.48 | **1.46** |
| RLBench [34] | Success Rate (%) ↑ | 36.4 | **55.5** | 24.3 | **61.7** |
| ALOHA [35] | | 60.7 | **63.3** | 12 | **55.3** |

## 4 Experiments

We evaluate VKT on Calvin [33], RLBench [34], and ALOHA [35]. Calvin and RLBench are standard benchmarks in language-conditioned multi-task manipulation learning. ALOHA is a bimanual manipulation environment. Further, we evaluate VKT in a subset of Open-X Embodiment [4] and real robot experiments. We aim to answer the following questions: (1) Can visual kinematics forecasting improve specialized agents in each environment? (2) Can VKT serve as a strong general agent in multiple environments? (3) Can VKT be efficiently trained with real-world demonstrations and solve manipulation tasks with real robots?

**Setup.** We adopt the setting of RT-X [5] for imitation learning from RGB images without depth. RLBench contains various task categories, such as pick-and-place, tool use, high-precision operations, screwing, tasks with visual occlusion, and long-term manipulation. RLBench provides five RGBD cameras. We use the RGB signal from the front, left, and right cameras and exclude the cameras installed on the shoulder and gripper of the robot because they do not take images of the robot. We choose 17 tasks based on camera visibility and the task categorization of Hiveformer [36]. We use 100 recorded trajectories per task for training, evaluate the trained agent on 25 independent trials, and report the average success rate. Calvin requires policies to accomplish multiple tasks sequentially in each episode. We use the training set and evaluation procedure of the Calvin D environment and report the average number of successful tasks per episode (average length) [33, 37]. ALOHA requires a high-frequency control of two robot arms through joint positions to solve tasks collaboratively. We chose the cube transfer task that requires picking up the cube and transferring it to the other arm. We follow the convention [37, 38] to report the metrics on Calvin and ALOHA with three independent runs of 1000 and 100 episodes, respectively. We report the success rate of RLBench

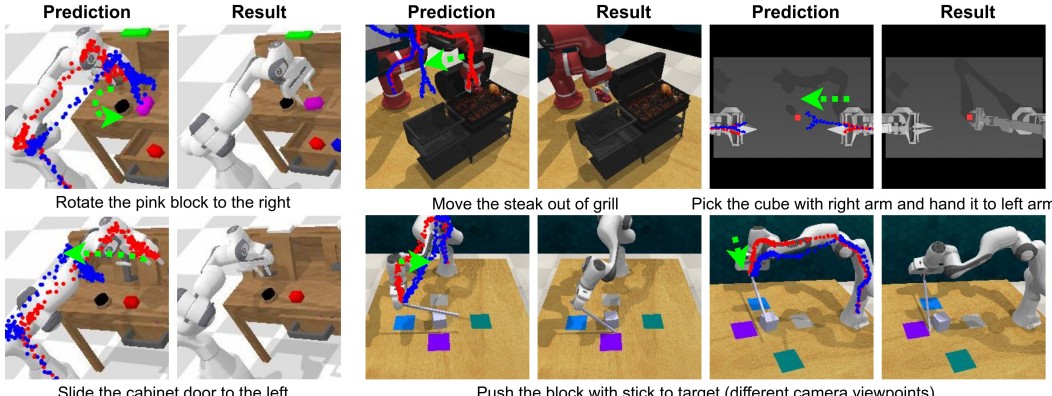

Figure 5: **Predicted visual kinematic chains returned by our VKT for different robots in Calvin (left), RLBench (right), and ALOHA (top right).** The predicted kinematic chain of the current frame is colored in red and the forecast one for the next time-step is in blue.

with five independent runs of 25 episodes [3]. Examples of Calvin, RLBench, and ALOHA tasks are shown in Figure 5. Additional visualizations are provided in the supplementary material.

**Implementation Details.** We use CLIP ViT-B/16 [32] as text and vision encoders. We resize all RGB inputs to $224 \times 224$. We use 8 multi-view dual attention blocks ($L = 8$) and predict visual kinematic chains for a horizon of 20 time-steps ($T = 20$) with 100 points ($N = 100$) at each step and set size $D$ to 512 in all experiments. RLBench has 3 camera viewpoints. ALOHA and Calvin have a single viewpoint each. During training, each minibatch consists of samples from all these three environments. Moreover, the three viewpoints of RLBench are randomly shuffled for every sample. The number of parameters is 114.8M in total and 32.5M without the ViT-B vision encoder. We adopt LoRA finetuning [39, 40] on the vision encoder. The original weights of the ViT-B vision encoder are kept frozen. The convolution head has only 96K parameters, less than 0.1% of the entire network. The structure of the convolution head is illustrated in Figure A4. We train the VKT for 50000 steps with a batch size of 96, which takes roughly 22 hours on a dual NVIDIA A6000 GPU server. The inference throughput is 0.02 secs/img on a NVIDIA 3060 GPU.

## 4.1 Learning from Multiple Environments

### 4.1.1 Simulation

**Baselines.** We compare the performance of our visual kinematics transformer (VKT) with a behavioral cloning transformer (BCT). The BCT is directly trained to predict the robot actions such as 6-DoF end-effector poses and joint values. In contrast, VKT is trained to forecast visual kinematic chains first, then project the kinematics tokens to robot actions using a convolution head with the backbone frozen, as shown in Figure 4. Note that the BCT shares the same neural architecture with VKT, including the convolution head for each specific environment. The specialized agent is trained on each environment separately. The general agent is trained using demonstrations from all environments. We use the 6-DoF end-effector pose and binary gripper states as the robot action space for both Calvin and RLBench, and the radian values of the 14 robot joints for ALOHA. We also compare the performance of our VKT with existing environment-specific methods. In Calvin, recent methods go beyond our imitation learning setup and adopt model learning for long-horizon latent planning. Therefore, we choose the goal-condition behavior cloning (GCBC) baseline from [41]. In RLBench, we choose PerAct [3] and retrain the model with the same camera setup as our VKT. In ALOHA, we choose the action chunking transformer (ACT) from the origin ALOHA paper[35].

**Results.** Table 1 and A3 summarize the comparison between VKT, BCT, and environment-specific methods. By forecasting the visual kinematic chains, VKT outperforms BCT in both specialized and general agents. This indicates that the kinematics tokens capture sufficient information about the robot's action, despite not seeing any low-level robot actions. When trained in all environments, the general BCT agent suffers a catastrophic performance drop (1.36 → 0.48, 36.4% → 24.3%, 55.3% → 12%). This indicates the challenge of policy learning from different action spaces. In

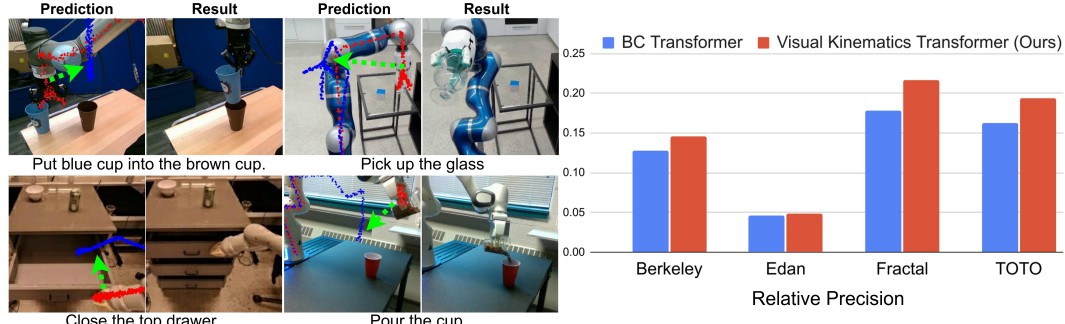

Figure 6: **Results on Open-X Embodiment.** Visual kinematic chains of different robots predicted by our VKT in Open-X Embodiment (left). The right figure compares the relative precision of BCT and VKT on the four selected datasets. BCT is randomly initialized and VKT is initialized from weights that are pretrained with visual kinematics forecasting. Relative precision is defined as an inverse of L1 error on robot actions, which is detailed in Section 4.1.2.

contrast, the general VKT agent has a much smaller performance drop in Calvin (1.58 → 1.46) and ALOHA (63.3% → 55.3%) but its performance increases (55.5% → 61.7%) in RLBench. This indicates that by learning actions in the image planes, training becomes more robust to distribution shifts. This is further verified by our real-robot experiments in Section 4.2.

**Failure Case Analysis.** We list the per-task success rates of RLBench and Calvin in Table A4 and A5. Overall, VKT outperforms BCT. We empirically find that VKT is more robust to changes in object layout because of the visually grounded action space. However, VKT has a lower success rate in the "turn tap" task. This task requires some specific end-effector rotations which involve large variations in the Euler angles of the end-effector but small changes in the visual kinematic chain from all camera viewpoints, as illustrated in the fourth row of Figure A5. Figure A5 summarizes the four categories of failure cases of VKT.

### 4.1.2 Open-X Embodiment

**Setup.** Open-X Embodiment (Open-X) [5] is a collection of video demonstrations of real-world manipulation tasks crowd-sourced from diverse environments. Since Open-X does not provide robot models such as URDF, and camera parameters are not available in most datasets, we select four datasets from Open-X and manually annotate the robot joint positions from 80 demonstration videos with 1,1359 frames. In future work, it is possible to apply robot keypoint detectors [42, 43] to extract the kinematic chain without URDFs. Using the annotated videos, we train a single VKT for the four datasets. Then, we train the network to predict low-level robot actions as in Figure 4. However, instead of freezing the VKT, we continue finetuning the entire network with the trained VKT as initial weights. For the BCT, we use the same training setup but randomly initialize the network.

**Results.** Figure 6 shows qualitative examples of the predicted kinematic chains (left) and compares the relative precision of BCT and VKT on the validation set (right). Let $\mathcal{L}_{vkt}$ and $\mathcal{L}_{bct}$ denote the L1 error of VKT and BCT, the relative precision of VKT is computed as $|\mathcal{L}_{vkt} - \mathcal{L}_{bct}|/\mathcal{L}_{vkt}$. Figure 6 shows that our VKT learns to predict kinematic chains from diverse real-world data, and visual kinematics forecasting can serve as a pre-training objective to improve imitation learning.

### 4.2 Real Robot Experiments

**Setup.** We evaluate our VKT in a real-world language-conditioned pick-and-place task. The task input includes an RGB image and text instructions such as "put the cup into the blue box". We adopt nine YCB objects such as balls, chips, and two boxes as place targets. Examples of this task and the forecasted kinematic chains are shown in Figure 7. We implement a scripted policy using a DINOv2-based object detector [44] to collect 90 demonstrations as training data, with only 5 demonstrations for each object-container pair. We automatically compute the ground-truth visual kinematic chain with the robot's URDF and the calibrated camera parameters. We use a Kuka LBR

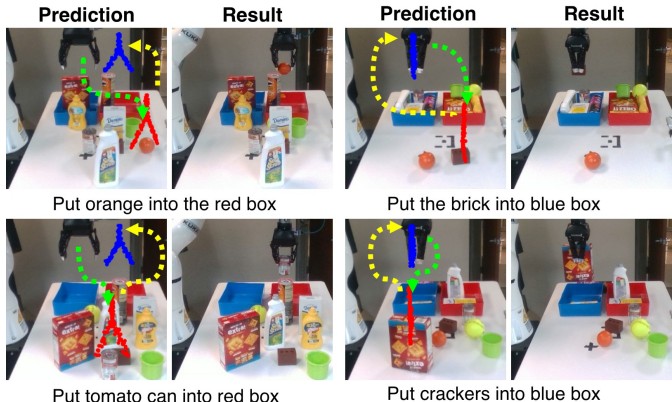

| Prediction | Result | Prediction | Result |
|---|---|---|---|
| Put orange into the red box | | Put the brick into blue box | |
| Put tomato can into red box | | Put crackers into blue box | |

Figure 7: Visual chains predicted by VKT in a language-conditioned pick-and-place task with a real robot. The chains of pick and place actions are in red and blue, respectively.

Table 2: Success rates of VKT for solving the real-world language-conditioned pick-and-place task.

| Object | BCT | VKT (Ours) |
|---|---|---|
| Overall (%) | 41.5 | **69.2** |
| Crackers | 3/7 | 4/7 |
| Cup | 3/7 | 4/7 |
| Ball | 3/8 | 7/8 |
| Chips | 1/7 | 3/7 |
| Brick | 5/8 | 6/8 |
| Tomato Can | 4/7 | 6/7 |
| Mustard | 0/7 | 4/7 |
| Sugar Box | 4/7 | 6/7 |
| Orange | 4/7 | 4/7 |

iiwa robot. For the convolution head, we train the model to directly predict the pick and the place poses instead of predicting the full trajectory for simplicity, and use MoveIt [45] for path generation.

**Results.** Table 2 shows the success rates of VKT in the real-world task. The overall success rate is the average of 65 independent runs with different object layouts. The BC Transformer (BCT) shares the same training and evaluation setup with VKT, but VKT outperforms BCT by a significant margin. Further, we discover that visual kinematic forecasting enables the use of 2D image augmentation for manipulation learning because the action space also resides in image planes, which is important to prevent overfitting in our experiment due to the small training set. We randomly augment the images through size scaling, translation, rotation, and perspective transform. In comparison, existing manipulation learning augmentations are applied in the 3D space [46], which requires depth that is less available than RGB data, and has fewer categories than 2D image augmentations [47].

**Details of analyses and ablation studies are included in Appendix A.1 due to space limit.**

## 5 Discussion and Future Possibilities

We have shown that visual kinematics forecasting of quasi-static robot movements can be a building block technique for developing general agents across diverse robot learning environments. In the following, we discuss the limitations and potential future directions.

**3D Kinematic Chain.** In Figure A5, VKT shows an inferior performance when the desired robot actions only produce small visual changes in the kinematic chain. This discrepancy is caused by the ambiguity of the 3D to 2D projection and prohibits VKT from more dexterous or precise tasks. One possible solution is to project the 2D kinematic chain back to a normalized 3D space with a virtual camera. This ensures the 3D space is consistent across different environments, provides the necessary precision in the 3D space, and still be compatible with our 2D kinematic chain method.

**General Visual Encoding of Robot Actions.** From an encoding perspective, the visual kinematic chain can be viewed as a visual encoding strategy for robot actions. The benefit of visual encoding is that it transforms robot learning tasks into vision tasks, where more generalized and robust solutions are often discovered. One possible future direction is to explore more visual encoding strategies with better learnability, higher precision, and reliable or even analytical translation into robot actions.

**Beyond Universal Action Representation.** Despite our advancements in unifying the action spaces across diverse environments, there is still a considerable gap in performance between our general agent VKT and those environment-specific state-of-the-art solutions. Different environments not only differ greatly in action spaces but more so in types of tasks and the required robotics skills. These skills are acquired separately by those environment-specific solutions. It is a potentially promising future direction to not just focus on universal action representation, but designing a universal learning architecture that can acquire various robot skills for diverse environments.

## Acknowledgement

This work is partially supported by NSF awards 1846043 and 2132972.

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

# A  Appendix

## A.1  Ablation Studies

Table A1: General agent's performance on forecasting the full kinematic chain vs. only the end-effector

| Configuration | | Calvin | RLBench | ALOHA |
|---|---|---|---|---|
| | | Avg. Length ↑ | Success Rate (%) ↑ | |
| BCT | | 0.48 | 24 | 12 |
| VKT (Ours) | Full Chain | **1.460** | **55.5** | **55.3** |
| | End-Effector | 1.24 | 54.1 | 28 |

Table A2: Effect of multiple viewpoints in RLBench

| Configuration | Success Rate |
|---|---|
| Single View | 26.8% |
| Multi View | 55.5% |

**Predicting the Full Chain versus Predicting Only the End-Effector.** We compare the effects of forecasting the full chain and only the end-effector in Table A1. Table A1 shows that forecasting the full chain improves the performance on all environments and especially the bimanual manipulation ALOHA benchmark. Moreover, despite the reduced capacity to represent rotations, end-effector VKT also delivers stronger performance as a general agent than BCT. This indicates that encoding robot actions visually in the image plane is the key to countering distribution shifts from different action spaces when building general agents across diverse environments. Furthermore, the end-effector predictions can be useful in special cases when robot models are unavailable. This suggests a broader applicability for visual kinematics forecasting.

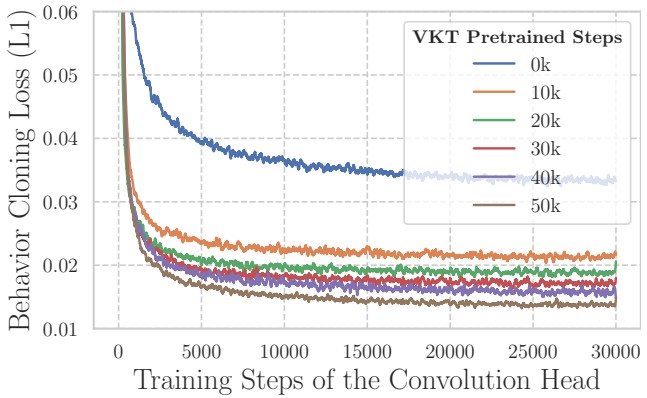

Figure A1: Behavior cloning Loss curves of the convolution head for Calvin. The VKT backbones are trained with different steps. We find a strong correlation between the VKT performance (represented by the pretrained steps) and the imitation learning convergence.

**VKT Performance and Imitation Learning Convergence.** We study the impacts of VKT performance on imitation learning convergence for the low-level robot actions in Figure A1. We train the VKT for a different number of steps. Next, we use the pretrained VKTs as backbones and train the convolution head for Calvin. Figure A1 demonstrates a strong correlation between the VKT pretrained steps and the convergence quality of imitation learning on robot actions.

**Earth Moving Distance versus Chamfer Distance.** Figure A2 compares the predicted visual kinematic chains using our proposed EMD and Chamfer Distance [48]. Unlike EMD, Chamfer Distance is not density-aware and only minimizes the distance between each point and its nearest neighbor. Figure A2 shows EMD produces fuzzier but more accurate kinematic chains, but Chamfer Distance produces sharper but sparser kinematic chains that lack structural integrity.

**Percentages of Training Data for the Convolution Head.** Figure A3 shows the average lengths on Calvin with the convolution heads trained on different data percentages. Similar levels of performance can be obtained with 5% data, or even 1% data at a minor performance drop. Calvin includes 34 tasks and 5123 episodes as the training set for $D \rightarrow D$ challenge. 1% data only leaves roughly 1

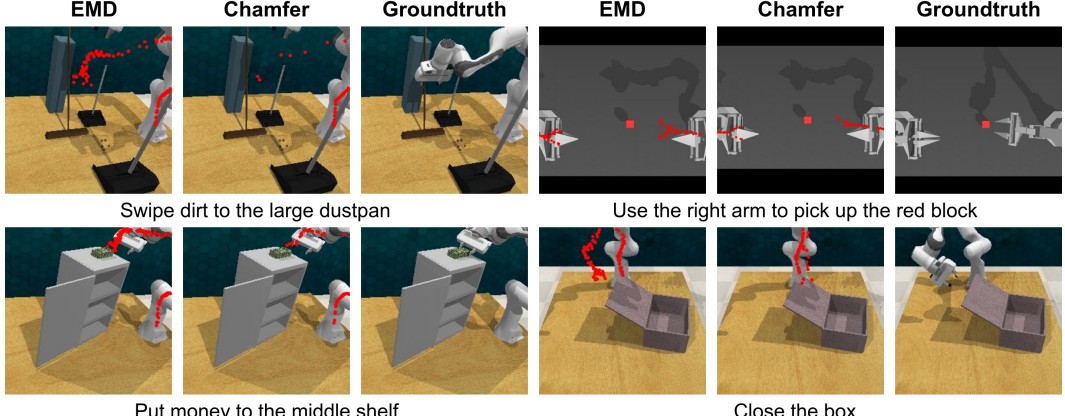

| EMD | Chamfer | Groundtruth | EMD | Chamfer | Groundtruth |

Swipe dirt to the large dustpan          Use the right arm to pick up the red block

Put money to the middle shelf          Close the box

Figure A2: **Comparison of Predicted Visual Kinematic Chain using EMD and Chamfer Distance [48].** The kinematic chains predicted by EMD are visually fuzzier but more accurately describe the structure of the robot arm. The chains predicted by Chamfer Distance lack structural integrity despite being visually sharper. Unlike EMD, Chamfer Distance is not density-aware and only minimizes the nearest neighbor distance for each point.

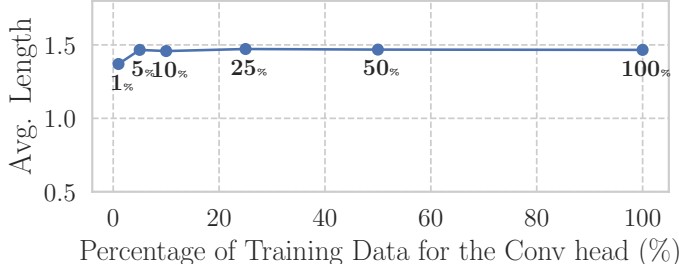

Figure A3: Average lengths on Calvin by training the convolution head with different data percentages. Calvin includes 34 tasks and 5123 episodes as the training set for $D \rightarrow D$ challenge. Similar levels of performance can be obtained with 5% data, or even 1% data at a minor performance drop.

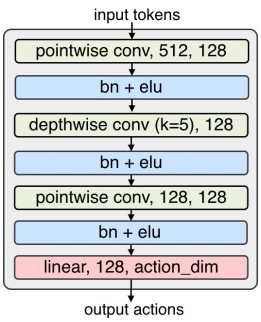

Figure A4: Architecture of the convolution head.

episode per task. This shows that kinematics tokens already encode robot action information. Only a small amount of data is needed to train a decoder to map kinematics tokens to actions.

**Multi-View versus Single-View Predictions.** We study the impacts of multi-view forecasting in Table A2. We remove the multi-view attention layers to create the single-view VKT. We average the low-level robot actions predicted from each view as the final prediction. Table A2 shows that VKT improves its 3D understanding significantly with our multi-view dual attention block by predicting consistent kinematic chains in different camera viewpoints.

## A.2 Additional Tables and Figures

Table A3: **Comparison of our VKT to environment-specific methods.** We show that our general agent VKT can outperform GCBC [41], and reach performance levels comparable to PerAct [3], two works published in 2020 and 2022, respectively. VKT also shows competitive performance against ACT [35], the SOTA method on ALOHA.

| Environment | Metric | Specialized Agent | | | VKT (Ours) | |
|---|---|---|---|---|---|---|
| | | GCBC [41] | PerAct [3] | ACT [35] | General | Specialized |
| Calvin [33] | Avg. Length ↑ | 1.11 | - | - | 1.46 | **1.58** |
| RLBench [34] | Success Rate (%) ↑ | - | 60.4 | - | **61.7** | 55.5 |
| ALOHA [35] | | - | - | **76** | 55.3 | 63.3 |

Table A4: **Performance on RLBench.** We report the success rates on each task and the average overall success rate.

| Agent | Method | Avg. Success | Put In Drawer | Reach Drag | Turn Tap | Slide Blocks | Open Drawer | Money In Safe | Place Wine | Sweep To Pan |
|---|---|---|---|---|---|---|---|---|---|---|
| Specialized | PerAct [3] | 60.4 | **60.4** | 60.0 | 64.8 | **52.2** | 68.0 | 34.8 | 10.0 | 46.0 |
| | BCT | 36.4 | 1.6 | 77.6 | 63.2 | 12.0 | 20.8 | 44.8 | 44.0 | 5.6 |
| | VKT (Ours) | 55.5 | 52.0 | **88.8** | 40.8 | 22.4 | 56.0 | 76.8 | 31.2 | 48.0 |
| General | BCT | 24.3 | 0.0 | 28.4 | **68.8** | 8.0 | 8.6 | 8.0 | 4.0 | 0.0 |
| | VKT (Ours) | **61.7** | 48.8 | 77.6 | 45.6 | 12.8 | **73.6** | **81.6** | **89.6** | **50.4** |
| | | Meet Off Grill | Phone On Base | Lid Off Pan | Close Microwave | Close Box | Ball In Hoop | Push Button | Lift Block | Close Jar |
| Specialized | PerAct [3] | 40.4 | 54.8 | 87.2 | 84.8 | 71.6 | 78.8 | **80.8** | **62.0** | **24.4** |
| | BCT | 32.0 | 37.6 | 47.2 | **93.6** | 84.0 | 33.6 | 21.6 | 0.0 | 0.0 |
| | VKT (Ours) | **41.6** | 68.0 | 92.0 | 64.0 | **93.6** | 85.6 | 64.0 | 8.8 | 10.4 |
| General | BCT | 21.6 | 0.0 | 52.0 | 80.0 | 76.8 | 36.4 | 20.0 | 0.0 | 0.0 |
| | VKT (Ours) | 33.6 | **72.8** | **98.4** | 92.0 | 84.8 | **93.6** | 69.6 | 11.2 | 12.8 |

Table A5: **Performance on Calvin.** Calvin requires agents to accomplish 5 tasks sequentially at each episode. We report the step-level success rate, average length and standard deviation of average length.

| Agent | Method | Avg. Length ↑ | Success rates at each task (%) | | | | |
|---|---|---|---|---|---|---|---|
| | | | 1 | 2 | 3 | 4 | 5 |
| Specialized | MCIL [49] | 0.41 | 34.4 | 5.8 | 1.1 | 0.2 | 0 |
| | GCBC [41] | 1.11 (0.3) | 64.7 | 28.4 | 12.2 | 4.9 | 1.3 |
| | BCT | 1.36 (0.06) | 63.2 | 39.6 | 18.6 | 9.7 | 4.9 |
| | VKT (Ours) | **1.58** (0.06) | **67.6** | **44.3** | **24.1** | **14.5** | **7.5** |
| General | BCT | 0.48 (0.1) | 36.4 | 8.6 | 2.6 | 0.4 | 0 |
| | VKT (Ours) | **1.46** (0.08) | **64.9** | **41.4** | **22.3** | **12.8** | **6.4** |

## Precision

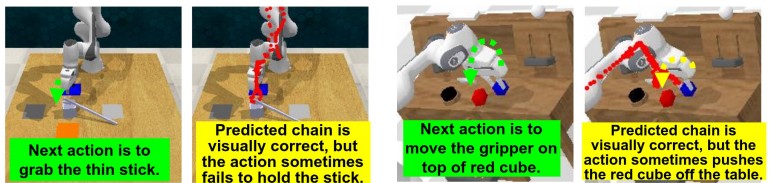

Drag the cube with a stick   Stack the blue cube on top of red cube

## Incorrect Kinematics Chain Prediction

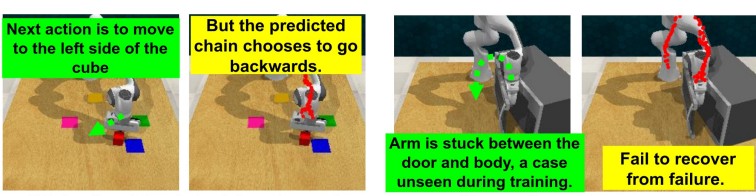

Slide the block to blue target   Close the microwave

## Ambiguity of Projection 3D to 2D

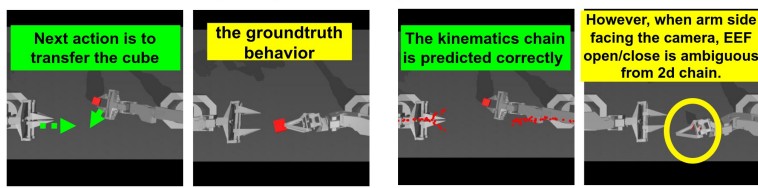

Pick up the red block with right arm, and transfer it to left arm

## Rotations with Tiny Visual Changes

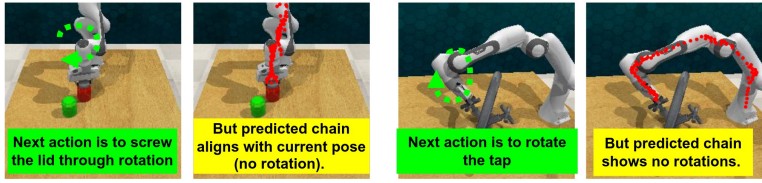

Close the red jar   Turn the left tap

Figure A5: **Failure Case Analysis.** The predicted kinematic chain of the next step is colored in red. The expected action is shown in a green arrow. The **first** row shows tasks: *drag with stick* and *stack blocks*. The kinematic chain is visually correct to represent the grabbing and stacking action. However, the predicted action sometimes lacks the necessary precision. The **second** row shows tasks: *push blocks* and *close microwave*, which requires spatial reasoning. However, the predicted chain represents the wrong actions. The **third** row shows the *cube transfer* task. The next action is to move two arms closer with the right arm grasping the cube. The predicted chain is visually reasonable. However, the gripper state is ambiguous when the right arm faces the camera from the side. When the right arm drops the cube, the task fails. The **fourth** row shows the tasks: *close jar* and *turn tap*, which require rotation actions with only small changes in the visual kinematic chain. The predicted chain does not show rotation actions.

