# OpenReview forum: "Scaling Manipulation Learning with Visual Kinematic Chain Prediction"
_robot-learning.org/CoRL/2024/Conference — CoRL 2024_

### Official Review · Reviewer_TuBj · 2024-07-11
**Intersting and novel idea for an universal action space for multi-embodiment robot learning with lacking experimental evaluation and too short limitation section..**

**Originality:** 3
**Technical Quality:** 3
**Clarity Of Presentation:** 4
**Potential Impact:** 3
**Recommendation:** 3
**Confidence:** 4

**Review:**

Strengths

- The proposed idea is novel and creative and tackles a relevant  problem in scaling robot learning with heterogeneous data consisting of various action spaces
- One big plus is the easy adaptability towards setups with multiple cameras and it also does not require depth
- several experiments in sim and real robot setups as well as general visualization on OXE show the performance of VKT to better scale than default BC Transformer
- Illustrative figures and website to demonstrate the proposed method

Weaknesses

- The method requires URDF files to get relevant kinematic information. In the OXE experiments, the authors already mention that they are not able to train on more data since this information is missing.
- the CALVIN results are very low compared to sota models such as MDT [1] for CALVIN D or 3DDA [2] for CALVIN ABC benchmarks. An average rollout length of 1 is not good on any of these benchmarks and does not provide any arguments for the proposed method. Further, it is not mentioned which dataset split the authors use for their experiments on CALVIN. In addition, no baselines are shown for RL-bench experiments. Adding some relevant 2D baselines would underline the effectiveness of the proposed method.
- the proposed method is not tested against other universal action spaces used in recent works such as: track points [3], video [4] and is only tested against default BC. These makes it hard to evaluate how well the proposed method compares against similar methods. Overall, relevant prior work is missing in the Intermediate Action Representation Section of the paper.
- No information about training time and inference speed
- the limitation section is not very detailed and is missing information


[1]: Reuss, Moritz, et al. "Multimodal Diffusion Transformer: Learning Versatile Behavior from Multimodal Goals." RSS 2024

[2]: Ke, Tsung-Wei, Nikolaos Gkanatsios, and Katerina Fragkiadaki. "3d diffuser actor: Policy diffusion with 3d scene representations." _arXiv preprint arXiv:2402.10885_ (2024).

[3]: Wen, Chuan, et al. "Any-point trajectory modeling for policy learning."  RSS 2024

[4]: Du, Yilun, et al. "Learning universal policies via text-guided video generation." _Advances in Neural Information Processing Systems_ 36 (2024).

**Quality Of The Limitations Section:**

2

**Questions For Rebuttal:**

- How would you try to solve the issue of missing URDF files to enable training VKT on OXE on more diverse data for future work?
- Can you add some relevant baselines mentioned in the weakness section?
- Why is the average performance of VKT on CALVIN so low compared to other methods?
- Did you ever try to combine your kinematic chain prediction with a separate more powerful low-level transformer policy as used in [1,2] and only use the kinematic chain as a visual conditioning for the low level policy? Similar how [3] does it with their track conditioning? While I understand the idea to have a single model for everything, I believe that the performance of the proposed method can be enhanced a lot with a better low level policy. In my experience a small action head is not enough for challenging benchmarks as used in the paper.


—-

Post rebuttal

The added experiments and the updated limitations section further strengthen the paper. While the performance gap to specialist models remain on some benchmarks, I believe it’s mostly related to architecture design decisions and are not related to the proposed method itself. Overall, I believe the paper has timely and relevant contributions for generalizable action spaces that are interesting for the wider robot learning community. Thus I changed to weak accept. For a strong accept the authors would have required better performance across benchmarks. However, I believe the paper is still timely and relevant.

**Robotics Focus:**

4

**Summary Of Paper:**

The paper introduces Visual Kinematic Chains as a universal action space for multi embodiment robot training. The main idea is to track the kinematic chain of the robot arm that is robot and end-effector agnostic and can be pre-trained on diverse robot datasets such as Embodiment. They use a transformer to track the kinematic chain and for domain adaption finetune a small action head. The proposed method is evaluated on several simulation and real robot environments.

**Summary Of Recommendation:**

Overall, I really like the creative and novel contributions of the paper and appreciate the evaluation across several benchmarks. However, missing relevant baselines and low performance on sim environments and the limitation section are open issues, that need to be addressed. Thus, I only recommend weak reject for now.

---

### Official Review · Reviewer_t5uW · 2024-07-21
**Review for VKT**

**Originality:** 2
**Technical Quality:** 2
**Clarity Of Presentation:** 4
**Potential Impact:** 2
**Recommendation:** 3
**Confidence:** 3

**Review:**

This paper addresses an important problem, and shows positive results with their method in certain cases. However, I have several concerns about the paper:

The paper includes an important ablation study to investigate whether there really is a benefit to predicting the motion of the entire kinematic chain of the robot vs just predicting the motion of the end-effector of the robot in pixel space. The results of this study show that the success rate for predicting the full chain is 55.5% and 54.1% for predicting the end-effector. It’s unclear for this table how many trials were used, but assuming 1000 trials were used, the p-value for this result is 0.5 which is not statistically significant. Therefore, the paper does not show any quantitative benefit to predicting the entire kinematic chain of the robot relative to the end-effector, which significantly weakens the paper given that the paper’s main emphasis is on predicting the full kinematic chain.

Additionally, the paper motivates the proposed approach by stating the need to “bridge the gap between different action spaces” in different datasets. However, for the simulation experiment considering the CALVIN and RL bench environments, the paper states that they both use the 6DOF end-effector pose as the action space. I don’t understand how this simulation study is able to study how well this method bridges the gap between different action spaces without having different action spaces. It seems instead like the main difference between the two environments is that the cameras are in different locations and the scenes and robots look different.

The paper shows significant improvement using the VKT model over the BCT model for the distribution shift between the two environments in the general agent case. This is fairly surprising given that predicting the full kinematic chain is not any better than predicting only the end-effector motion and only predicting the end-effector motion (which performs as well as VKT with kinematic chain prediction) does not capture rotations. How can a method that cannot capture end-effector rotations perform better than one that can? Additionally, while both VKT and BCT have the same architecture, in the current implementation, only VKT gets access to a specialized head that has been finetuned on each environment separately. This specialized head has learned to convert actions in the camera frame (i.e. pixels in the image plane) to the robot’s frame. Therefore, an important baseline to evaluate VKT vs BCT would be to have BCT predict actions in the camera frame and then convert those actions into the frame of the robot (which is straightforward to do with known camera parameters which is an assumption VKT makes) before passing them to the robot.

The success length metric is confusing. The paper cites reference [28] as an example of reporting number of successful steps per episode, but I was not able to find where this metric is reported in this paper. If the success length is the average number of successful steps per episode, why is the average close to 1 or less than 1 for all methods? This seems very low. What is the length of the demonstration trajectories? When reporting this metric, the standard deviation of the average length should also be reported to evaluate statistical significance of the results.

Each results table needs to clearly indicate how many evaluation trials were used for each experiment. The current paper has conflicting information on this topic. For example, in Table 2, the caption states that “success rates are measured from five independent runs.” However, the success rates reported in Table 2 have three significant digits, which is impossible to obtain from 5 trials. Not knowing how many trials were actually used in each experiment makes it difficult to properly evaluate the statistical significance of the results.

The bar chart on Figure 6 is difficult to interpret. The text needs to explain this plot in more detail.

**Quality Of The Limitations Section:**

3

**Questions For Rebuttal:**

Lines 215 and 216 refer to 2D image augmentation for manipulation learning. What augmentations are used?

**Robotics Focus:**

4

**Summary Of Paper:**

Action spaces are frequently not aligned between different robotics datasets, and this paper proposes a method to address this problem. Instead of training a model to predict the 6DOF pose of the end-effector, they train a model to predict, in pixel space, the motion of the kinematic chain of the robot. This kinematic chain can be obtained using the robot’s URDF and a calibrated camera. The performance of this method is evaluated both in simulation and on real hardware and is compared to a BC transformer baseline that predicts the robot’s end-effector pose directly.

**Summary Of Recommendation:**

The paper addresses an important problem and shows some positive results. The emphasis on predicting the entire visual kinematic chain despite the negative results of the ablation study is very confusing. The paper needs some clarification on some of the results as described in the main review.

---

### Official Review · Reviewer_Lf1f · 2024-07-27
**A good paper that might needs more clarification and analysis**

**Originality:** 3
**Technical Quality:** 3
**Clarity Of Presentation:** 3
**Potential Impact:** 3
**Recommendation:** 3
**Confidence:** 4

**Review:**

- strength:
  - The proposed representation and method are novel and interesting
  - There are sufficient experiments demonstrating the effectiveness of this method
  - The writing is clear and easy to read overall.
- weakness:
  - Potential unfair comparison. The authors compare the specialized agent and the general agent in the paper. For the general agent, the training details are not quite clear. My current assumption is:
    - General VKT: train VKT on all tasks and train the tiny head with one task (as described in L135).
    - General BCT: train BCT on all tasks.

    If it is true, I think the general agent comparison is unfair. Because BCT can only have one model to consume all data, while VKT can train separate tiny heads for each task. If my assumption is wrong, please also clarify it.
  - More analysis is needed for experiments. Table 2 lists the success rate of different tasks. However, the analysis is not clear. It could be more insightful if the authors could summarize the failure mode of BCT and VKT. For example, one failure pattern of VKT is the motion requiring rotation. There are also some interesting data. For example, BCT performs drastically worse on "put in drawer". Analyzing why VKT performs better could be interesting.
  - Tasks it can accomplish are still preliminary, such as pick-and-place and pour. Extending this framework to more dexterous tasks could be more impressive.
- comments:
  - Table 2: The layout can be a bit confusing at first glance. I thought it was comparing "Specialized BCT" vs. "Agent VKT" or "General BCT" vs. "Agent VKT". Maybe "Specialized Agent" and "General Agent" can be summarized as "Specialized" and "General". Then they should be aligned vertically.
  - All image captions and table captions will be easier to notice and read by adding an image or table summary in **bold** fonts. Now captions look a bit mixed with the main text

**Quality Of The Limitations Section:**

3

**Questions For Rebuttal:**

- Has the authors tried different losses for VKT? Is earth-moving loss significantly better than other loss, like Chamfer distance?
- Is N (the number of predicted points) a fixed number?
- For specific environment deployment, how much data is needed to train the tiny head?
- How scalable this method is with respect to the number of camera viewpoints? For example, why only three viewpoints out of five are chosen in RLBench?
- Is it possible to raise visual kinematics to 3D to resolve the rotation motion problem?

**Robotics Focus:**

4

**Summary Of Paper:**

This paper proposes the visual kinematics chain as a novel action representation to enable manipulation policy training on diverse data with various embodiment and environments. The authors also introduce the Visual Kinematics Transformer (VKT) to predict kinematics structure and use it to generate actions for downstream tasks.

**Summary Of Recommendation:**

This paper is overall a good paper and novel. If authors could clarify their experiment setting and add more experiment analysis, I lean to accept it.

---

### Author Rebuttal · Authors · 2024-08-13

We are greatly thankful to the reviewers for the feedback and acknowledgment of our work. We have addressed the concerns raised and uploaded the revised paper in the rebuttal file (changes are colored in blue). We briefly summarize the revisions in the following.

1. **New Environment: ALOHA bimanual manipulation.** We show that our general agent VKT achieves competitive performance on ALOHA, in addition to RLBench and Calvin (Table 1). Unlike RLBench and Calvin which use EEF poses, the action space of ALOHA consists of the radians of 14 joint positions of two arms. This demonstrates the capability of the proposed visual kinematics forecasting in bridging various action spaces. Further, we demonstrate a significant performance gain that results from employing full-chain predictions compared to end-effector predictions in the ALOHA bimanual manipulation tasks (Table A1).
2. **Standard Baselines.** We add several standard environment-specific solutions as baselines to VKT (Table A3), and show the competitive performance of VKT as a general agent.
3. **More Comprehensive Analysis.** We add studies on (1) failure cases, (2) EMD versus Chamfer Distance, (3) different percentages of training data for the convolution head, (4) the correlation between VKT performance and imitation learning convergence (Appendix A.1).
4. **Improved Related Work Discussion.** We improve the related work section by addressing the main differences between our VKT and the recent work on universal action spaces, such as point tracks and videos (Lines 88-96).
5. **In-depth Discussion on Limitations.** We add an in-depth discussion on the limitations and future directions of our work (Lines 228-229, 259-280), including how to raise the kinematic chain predictions to 3D space for more dexterous and precise tasks, how to handle missing URDFs, visual encoding of robot action as a general strategy to transform robot learning to vision tasks, and go beyond universal action representation to finding a universal learning architecture that can acquire various robot skills for diverse environments and task genres.
6. **Clarity.** We clarify our experiment settings and implementation details for the related concerns.

Detailed responses can be found in the following individual comments.

---

### Decision · Program_Chairs · 2024-09-04

**Decision:**

Accept

**Comment:**

This paper introduces the Visual Kinematic Chain as a more general action space to scale up learning in robotic manipulation. The reviewers all recognize the novelty and importance of this approach.

Before the rebuttal, the reviewers raised several concerns. The experimental analysis is criticized for unfair comparisons and lack of detailed analysis. The paper does not clearly show the quantitative benefits of predicting the entire kinematic chain versus just the end-effector. Additionally, the claim of bridging different action spaces is not well-supported, and the reason for VKT's improvement over BCT is unclear.

The evaluation methods and metrics also need clarification, especially the success length metric. Important details are missing, and the reliance on URDF files might limit generalizability. Comparisons with other action spaces like track points and video are insufficient.


During the rebuttal, the authors did an excellent job addressing the reviewers’ concerns. As a result, all reviewers now rate the paper as a Weak Accept. However, Reviewer t5uW still has concerns regarding the explanation of Figure 6 and the claims about performance improvements. Nonetheless, the reviewer acknowledges that the overall idea is interesting and valuable for the field. I strongly encourage the authors to consider the reviewers’ feedback both during and after the rebuttal to strengthen their paper.

I’m including Reviewer t5uW’s comment from the discussion below.

========

Thank you to the authors for their response.

One of my main concerns previously was the ablation study in Table A1 which previously showed no improvement for forecasting the full kinematic chain instead of only the end-effector. I stated that “the success rate for predicting the full chain is 55.5% and 54.1% for predicting the end-effector. It’s unclear for this table how many trials were used, but assuming 1000 trials were used, the p-value for this result is 0.5 which is not statistically significant.” I’m also not convinced that for the other environment a difference in average length of 1.24 vs 1.46 is a meaningful improvement. The authors need to perform a statistical comparison, or at the very least, provide a measure of variance of their reported performance. Despite this, the authors kept the claim in their revised draft that “forecasting the full chain improves performance on all environments” which I strongly disagree with given the p-value analysis I provided. The authors should remove the sentence in lines 411-413. I appreciate the new results on the ALOHA environment that show a meaningful improvement in performance for predicting the full kinematic chain. However, this means that predicting the full kinematic chain only leads to improvement in 1 out of the 3 tasks studied and no explanation was provided on why that is. The paper would be stronger if more analysis was provided to explain when and why predicting the full kinematic chain actually helps.

The authors’ response to my other concerns in W3 were helpful and I agree with their comments.

My other major concern are the results presented in Figure 6. I asked for more explanation from the authors for this figure, but the additional explanation provided merely repeated what was already said. In its current state, Figure 6 does not make sense. The text says that Figure 6 shows that VKT can predict kinematic chains from real world data. But all Figure 6 shows is a comparison to the BCT baseline.

The “relative precision” metric does not make sense — according to 4.1.2, it’s a relative comparison between two methods. By this definition, VKT and BCT could both be equally terrible. No data is provided to evaluate the absolute performance of either method, therefore a comparison between them is not helpful.

Also, if relative precision is indeed defined as in 4.1.2, it doesn’t make sense to show BCT and VKT as separate bars. Isn’t the whole point of the relative precision metric to produce a single scalar comparing the two methods?

The text in the authors’ rebuttal also does not make sense to me.

> “Figure 6 compares the relative precision (an inverse of L1 error on regression of robot actions, explained in lines 234-235) of BCT and VKT on validation splits of the selected datasets.”

- “relative precision” as defined in 4.1.2 is not the inverse of L1 error.

> “It demonstrates that visual kinematics forecasting, as a pre-training objective, contributes to the accuracy of action regression.”

- The authors do not show this. Are they trying to argue VKT is better than BCT? The actual paper text does not make this claim, and the figure does not support it.

I am clearly still missing what the authors are trying to show. The current figure and text in 4.1.2 do not make sense together even after the authors attempted to clarify, in either the rebuttal or the text. If the goal is for the authors to show that VKT can produce kinematic chains, why do they not simply report L_vkt?

This is unfortunate because Figure 6 is fairly incidental to the overall story the authors are telling. I am fine with accepting this paper if the authors can either explain Figure 6 in a coherent fashion or simply delete it from the paper.

I am changing my evaluation to a weak accept because I think the ideas proposed in the paper are interesting and valuable for the field and the ALOHA results have strengthened the paper, but given my remaining concerns on the quality of the execution of the paper, I would be very hesitant to publish this paper unless

(1) the authors can coherently explain Figure 6 (or simply delete it)

(2) the authors remove the claim that a success rate of 55.5 versus 54.1 is meaningful in Table A1 (lines 411-413). It is simply not.